# CRISPR-Cas9 Direct Fusions for Improved Genome Editing via Enhanced Homologous Recombination

**DOI:** 10.3390/ijms241914701

**Published:** 2023-09-28

**Authors:** Tahmina Tabassum, Giovanni Pietrogrande, Michael Healy, Ernst J. Wolvetang

**Affiliations:** 1Australian Institute for Bioengineering and Nanotechnology, The University of Queensland, St. Lucia, Brisbane, QLD 4072, Australia; t.tabassum@uq.edu.au (T.T.); g.pietrogrande@uq.edu.au (G.P.); 2Institute for Molecular Biosciences, The University of Queensland, St. Lucia, Brisbane, QLD 4072, Australia; m.healy@imb.uq.edu.au

**Keywords:** CRISPR-Cas9, DNA repair, homologous recombination, gene editing

## Abstract

DNA repair in mammalian cells involves the coordinated action of a range of complex cellular repair machinery. Our understanding of these DNA repair processes has advanced to the extent that they can be leveraged to improve the efficacy and precision of Cas9-assisted genome editing tools. Here, we review how the fusion of CRISPR-Cas9 to functional domains of proteins that directly or indirectly impact the DNA repair process can enhance genome editing. Such studies have allowed the development of diverse technologies that promote efficient gene knock-in for safer genome engineering practices.

## 1. Introduction

Gene editing is a cutting-edge technology that is rapidly reshaping biotechnology, medicine, and agriculture disciplines. Precise alteration of genetic makeup requires the introduction of DNA lesions at a region of interest and exploits the DNA damage response and homology-driven repair mechanisms. DNA is prone to daily damage from various physiological and pathological factors [1], resulting in DNA double-stranded break (DSB) or single-stranded break (SSB or nick) that may trigger genomic reshuffling if left unrepaired or when improperly repaired [2]. These events can then trigger downstream processes, such as carcinogenesis or programmed cell death [3]. To sustain genomic integrity, a network of repair mechanisms has evolved, and their activation is dictated by the type of DNA damage caused by either endogenous or exogenous stress. Gene editing techniques harness the power of this intrinsic repair network to rewrite the DNA. The four main editing platforms include mega-nucleases, zinc finger nucleases (ZFNs), transcription activator-like effector nucleases (TALENs), and clustered regularly interspaced short palindromic repeats (CRISPR). Natural mega-nucleases trigger DNA damage but require unique recognition sequences for action, which makes it strenuous to find target region-specific endonucleases [4]. Efforts to re-engineer nucleases led to the development of alternatives, such as ZFNs and TALENs, where a DNA binding structure is fused to the cleavage domain of the FokI restriction enzyme. This greatly improved gene editing in human cells and animal models and thereby facilitated the therapeutic application of gene editing [5,6,7,8]. However, feasibility issues remain unresolved as these artificial nucleases, in addition to random off-target mutagenesis, require protein engineering for every change in the target sequence of interest, making the entire process laborious and expensive [9]. Packaging and delivery of mega-nucleases is also difficult, further limiting in vivo applications [7]. On the other hand, CRISPR technology has a very significant advantage over such editing modalities as it overcomes the need for protein engineering for every new target site, making it easily reprogrammable [4]. However, since CRISPR generates nonspecific DSBs that can introduce spurious indels, major safety concerns remain regarding the mutagenic potential of this technique.

Interestingly, most DNA-altering technology has been derived from the bacterial antiviral immune system, including the first restriction enzyme (HindIII) involved in defence against bacteriophages [10]. Similarly, CRISPR-Cas is a bacterial adaptive immune system discovered serendipitously while studying the effects of environmental salinity on haloarchaea [11]. The mechanism of action involves building molecular memory by incorporating bacteriophage genetic elements in the bacterial genome as repeated spacer sequences during the first bacteriophage infection [12]. The spacer sequences are transcribed to CRISPR RNA to guide Cas (endonuclease encoded by CRISPR-associated genes) proteins that destroy bacteriophage upon reinfection by the same strain [11]. The ribonucleoprotein complex interacts with recognised foreign DNA through the complementary base pairing of CRISPR RNA to a target site. The bacterial immune system has been an excellent source for discovering numerous molecular scissors, and continual research into this unique defence mechanism may potentially elucidate further applications in genetic engineering field.

The CRISPR-Cas system is broadly classified into Class 1 and Class 2 based on Cas effector protein structure and function and is further subdivided into types and subtypes [13]. The two classes include interaction with multiple proteins and single-protein effectors, respectively. The Class 2 Type II CRISPR-Cas9 system is most widely studied and has been applied in both translational and fundamental research [14]. CRISPR-Cas9 derived from *Streptococcus pyogenes* (referred to as Cas9 from here onwards) is guided to the site of action by guide RNA (gRNA) and, upon recognition of protospacer adjacent motif (PAM) sequence (5′-NGG-3′), the nuclease domains RuvC and HNH mediate endo-nucleolytic cleavage of the gRNA complementary strand, thus inducing DSBs [14]. Further variations, called nCas9, have been obtained by mutagenesis at either of these catalytic domains and produce nick instead of DSB at the targeted site. Mutation of the RuvC domain (Cas9-D10A) causes a nick in the strand complementary to the gRNA sequence, while HNH-mutant (Cas9-H840A) cuts the PAM containing the non-target strand of the DNA [15]. Inactivation of both nuclease domains of Cas9 endonuclease creates a catalytically defective Cas9 (dCas9) that retains its DNA binding ability guided by the gRNA. dCas9 has been extensively adopted for targeted gene regulation by directing proteins to specific sites of the genome through fusion at the N- or C-terminal. Such direct fusions of regulatory proteins to dCas9 have found broad application in gene editing technology including base editors such as ABEs, epigenome editors, such as dCas9-HDACs, transcriptional activators, such as dCas9-VPR and transcriptional repressors, such as dCas9-KRAB [16,17,18,19]. In addition, dCas9 has found use in long-term fluorescence imaging by fusing dCas9 to a protein scaffold called SunTag, which can recruit multiple copies of engineered antibodies fused to fluorophores for precise live imaging [20]. Detailed discussion on these applications is beyond the scope of this review as we strictly focus on Cas9 fusion proteins involved in enhancing genome editing accuracy.

## 2. High Fidelity DNA Repair

DSBs are widely studied as this type of DNA damage can cause mutations, chromosomal translocation, and cancer formation [21]. DSB repair includes two distinct cell-autonomous pathways: Non-Homologous End Joining (NHEJ) (Figure 1(Ai)) and Homologous Recombination (HR) (Figure 1(Bii)). NHEJ promotes direct ligation of broken strand ends, thus introducing random indels and frameshifts [22], whilst HR promotes accurate repair by utilising a homologous sequence as a template to maintain high fidelity [2]. Here, we mainly discuss homologous recombination as it is the key pathway for precise gene knock-in, which is the focus of this review. During HR, the DSB is first recognised by the MRN complex (MRE11/RAD50/NBS1) aided by ATM kinase, which opens the chromatin structure and causes initiation of the DNA-damage response. The MRN complex initiates DNA 5′-3′ resection in cooperation with CtBP-interacting protein (CtIP) nuclease. ExoI and BLM/DNA2 helicases form DNA 3′ overhangs, which are stabilised by replicative protein A (RPA). Rad51 loading on the stabilised overhang replaces RPA and forms the Rad51 nucleoprotein filament, which promotes homologous pairing through microhomology scanning and facilitates strand invasion of the intact homologous sequence via Rad54 catalysis [23]. Strand invasion by 3′-ssDNA primes DNA synthesis, forming a displacing loop, generating holiday junctions to be resolved by endonucleolytic cleavage for strand exchange. Double holiday junctions are resolved by topoisomerase III, while unresolved holiday junctions cause break-induced replication (BIR) or synthesis-dependent strand annealing (SDSA). As such, the DNA damage response relies on the coordinated action of multiple intrinsic cellular components. Moreover, post-translational modifications, such as phosphorylation, ubiquitination and SUMOylation, are necessary for the initiation and progression of HR [24] and their targeted modulation may prove to be a promising strategy to boost the rate of gene editing events. For example, UHRF1 plays an important role in HR initiation through ubiquitination of pro-NHEJ proteins [25] and PIAS4 is known to improve DSB repair accuracy through SUMOylation of multiple DNA repair factors [26,27]. Targeted localisation of such proteins involved in upstream regulation of HR at break site may improve gene editing rates.

Single-strand DNA nicks are the most frequently occurring DNA damage and, therefore, need to be continuously repaired to maintain genomic stability. Nicks are usually repaired rapidly by the base excision repair (BER) mechanism (Figure 1(Bi)) or by HR when converted to DSBs [28]. HR initiation at nicks can also occur without DSB formation [29], and this was hypothesised to involve a RAD51-dependant mechanism for the dsDNA template (Figure 1(Bii)) and a RAD51/BRCA2-independent mechanism for ssDNA or nicked dsDNA donor [30]. However, the incidence of HR at nicks without DSB formation is exceedingly low, and the molecular mechanism currently lacks clarity, but it is thought to be different to HR at DSBs in terms of the associated proteins. Although the occurrence of such repair is low, it is of paramount importance to better understand this process as it would pave the way for safer DNA editing systems. The low HR efficiency at nicks could be explained by a competitive interplay between BER and HR mechanisms; however, to date, we are still at the stage of speculation as no report has definitively shown this relationship. 

## 3. Improving Precise Gene Editing

Over the past decade, CRISPR-Cas9 has found widespread application in loss-of-function mutations, but precise genetic engineering for gene correction or gene replacement therapies has lagged behind. In vivo correction using CRISPR-Cas9 to replace genetic mutations by HR is highly challenging, and very few studies have managed to achieve this [31,32]. Ex vivo applications in monogenic disorders are more common, but large on-target indels may occur, which requires extensive screening and quality control of edited cells, making the process time-consuming and costly [33]. To overcome these limitations, an increasing number of studies are probing the fusion of various proteins and small molecules to Cas9 in an effort to favour DNA repair by HR over the more prevalent NHEJ [34]. Here, we discuss how cell cycle control, localisation of regulatory proteins in HR, epigenetic modification, and local donor saturation can favour HR in gene knock-in studies using Cas9 direct fusion variants (Table 1).

### 3.1. Cell Cycle Modulation

The cell cycle is generally divided into G1 (cell growth), S (DNA replication), G2 (preparation for mitosis) and M (cell division) phases [51]. The incidence of the DSB repair pathways is cell cycle-dependent, and the relative prevalence of NHEJ over HR during the cell cycle is a major limitation to the adoption of Cas9 for gene therapy purposes, as numerous studies associated Cas9 with the widespread introduction of on-target and off-target mutations [52,53]. NHEJ is active throughout the cell cycle but is highly favoured in the G1 and S phases [54]. On the other hand, HR is restricted to late S and G2 phases only as after DNA replication in the S phase, two copies of chromosomes are available for division. Therefore, the cell cycle phase plays a vital role in DNA repair response and thus impacts gene editing rates.

The concurrent activity of NHEJ and HDR is regulated by a competition mechanism between the two pathways; therefore, the promotion of one is possible by limiting the other [55]. Small molecules, such as nocodazole and SCR7, are widely used to arrest cells in S and G2 phases and thus limit NHEJ rates [55]. Suppressing regulators of NHEJ, such as PARP1 inhibitors, call forth global NHEJ inhibition but is accompanied by cytotoxicity [56]. Chemical synchronisation efforts have revealed variable precise editing efficacies across different cell types and significantly reduced effects in primary cell models [37]. In addition, chemical treatment usually requires laborious optimisation for each cell type and often affects cell fitness and viability [57]. To overcome these limitations, an optimal strategy is temporary inhibition of NHEJ only at the local molecular level. Colocalisation of Cas9 with a dominant negative version of the pro-NHEJ protein 53BPI (DN1S) limits NHEJ at induced break sites and increases HR frequency by 1.5–3-fold [38]. However, robust experimentation on different cell lines and loci revealed cell type bias, editing site bias, and increased cell cytotoxicity by 5–10% [38]. Comparative analysis between NHEJ inhibitory small molecule treatments and Cas9-fusions can be carried out to test for HR to indel ratios and cell cytotoxicity to identify the best-suited method to promote HR through NHEJ inhibition. It is important to identify other proteins that interfere with NHEJ regulatory proteins and explore their effect on the choice of repair when colocalised at DSB with Cas9. 

An alternative strategy for temporal modulation of gene editing events over the cell cycle can be achieved through the N-terminal fusion of Cas9 to truncated Geminin protein (Cas9-GE), thus restricting Cas9 activity to the G2/S phase [35]. Geminin is a direct substrate of E3 ubiquitin ligase complex APC/Cdh1 and is actively ubiquitinated in the M/G1 phase. Fusion to Cas9 promotes degradation of the chimeric protein during the M/G1 phase, restricting DSBs to HR-predominated phases only. This approach increases knock-in efficiency by 1.5–2-fold across cell models, and it has proven effective in cells not affected by small molecule treatments [35,36,37]. A recent study reveals Cas9-GE co-transfection with anti-CRISPR proteins (AcrIIA4/AcrIIA5) fused to the chromatin licensing and DNA replication factor 1 (Cdt1) ubiquitination domain synergistically improved HR rate (1.6–2.3-fold) and reduced indels by approximately 80% [58]. Anti-CRISPR-Cdt1 fusion protein inhibits Cas9 that escapes Geminin-associated proteolysis during M/G1, and this inhibition is removed upon ubiquitination of Cdt1 fusion protein during the G2/S phase [58]. Leveraging anti-CRISPR proteins to improve spatiotemporal control of Cas9 function is a promising avenue for reducing the mutagenic potential of CRISPR-based editing. 

### 3.2. Colocalisation of HR Effectors

Fusion of key regulatory proteins of the HR pathway to Cas9 can enhance CRIPSR-induced transgene insertion. To date, multiple proteins from the HR cascade have been fused to Cas9, either alone or in combinations, to study whether their local recruitment and enrichment would impact HR rates (Table 1). Most fusions with mediators of the MRN complex displayed higher HR to indel ratios, confirming the utility of Cas9 fusion with HR effector proteins as more precise genome editing tools that enhance HR at DSB [34]. Fusion of the DNA resection exonuclease CtIP to Cas9 forces its localisation to break site and improves gene knock-in efficiency at similar rates to Cas9-GE in human fibroblasts while also reducing indel frequencies [37,39] with no obvious negative impact on cell fitness. However, Cas9-CtIP fused to Geminin did not synergistically increase the efficiency of HR. One possible explanation is that Cas9-CtIP activity is confined to the G2/S phase, potentially limiting additional enhancement of HR. This observation is intriguing as it suggests that augmenting one mechanism could plateau in terms of increasing recombination efficiency. Consequently, boosting multiple elements within the same pathway could lead to redundancy rather than an amplified effect. Therefore, careful pathway selection becomes of the essence to ensure non-overlapping interactions and instead foster synergy between fusion strategies [35]. Consistent with this concept, the recruitment of two molecules of CtIP to Cas9 using MS2 tagging, an approach in which gRNA loops include an MS2 phage aptamer and CtIP is fused to MS2 coat protein for interaction, increased the HR to indel ratio. However, the recruitment of multiple CtIP motifs through a Cas9-SunTag array was not associated with a significant improvement in HR [41]. It was hypothesised that this ineffectiveness could be due to oligomerisation of multiple CtIP peptides or steric hindrance caused by its relatively large size, but no further study was developed to attest to these conjectures. We speculate that the significantly lower activity of Cas9-Suntag-CtIP compared to Cas9-CtIP and Cas9-MS2-CtIP may be due to either flexibility of the SunTag array, which reduces retention time of CtIP at the target site or recruitment of CtIP to a region on the array positioned further from target site. Identifying the minimal functional domain of HR proteins and recruiting multiple co-operative smaller HR protein domains that do not oligomerise to the Cas9-SunTag scaffold is an alternative that clearly warrants further research. A seminal study from Charpentier et al. [39] provided the foundation for numerous subsequent studies that aimed to enhance genome editing efficiency through localisation of key factors in the HR pathway. Fusion of a dominant negative mutant of E3 ligase Ring Finger 168 (dnRNF168) lacking the RING domain to Cas9-CtIP significantly decreased indel formation [59]. RNF168 recruits NHEJ regulatory protein 53BPI to DSB through the ubiquitination of surrounding H2A histones [60]. Carusillo et al. demonstrated safer editing of human primary cells using Cas9-CtIP-dnRNF168 by dual action of HR enhancement by CtIP and concomitant NHEJ inhibition by the dnRNF168 domain [59]. 

An alternative strategy targets the 5′-3′ end resection step in the HR cascade through N-terminal coupling of Cas9 with truncated Exo1, an enzyme involved in exonuclease activity during resection [44]. This approach increased knock-in rates while decreasing p53 pathway-mediated cytotoxicity by up to four-fold in mammalian cell models. However, this study restricted its scope to HR enrichment with a limited focus on the mutagenic potential of the tools. In addition, inconsistency in HR improvement across different cell types and gene loci remains, suggesting the existence of underlying mechanisms is still not fully understood. 

Rad51 loading after end resection is a critical step in the HR cascade that leads to the formation of nucleoprotein filaments involved in homology search (Figure 1(Aii)). Enriching Rad51 concentration in the DNA repair microenvironment can promote homology-directed repair, as demonstrated by Song et al. by the use of RS-1, a chemical agonist of Rad51 [61]. Previous efforts of coupling Rad51 to nCas9 have shown highly precise editing rates at nicks, but results were not consistent amongst different cell types [42]. Fusion of Cas9 to protein domains that mediate Rad51 recruitment turned out to be a more effective strategy. BRCA2 plays a key role in recruiting HR regulatory factors, including Rad51, to the site of action. Ma et al. showed that fusing a small motif of BRCA2 called Brex-27 to Cas9 (MiCas9) recruits Rad51 molecules and provides the highest HR rates compared to all other HR protein fusion combinations tested thus far, and results are consistent at different target loci across different cell models [43]. In addition, this strategy displays lower indel incidence at on-target and predicted off-target regions without impeding precise editing efficiency, making MiCas9 an attractive tool for safer and more efficient precise editing. A recent study elucidated that Rad51 localisation at the DSB site has applicability in homozygous gene conversion, doubling homozygous gene insertion efficiency in mouse models compared to Cas9 control [62]. This is of particular interest in improving gene corrective therapies for several genetic disease conditions as this repair system utilises its own wildtype allele for homozygous knock-in, bypassing the need for an exogenous template. This study provided a proof of concept for safer and more efficient editing via Cas9 fusions by demonstrating that Rad51 localisation can significantly enhance interhomolog repair in the mouse embryo. 

Another study reported that the fusion of Cas9 to DNA polymerase delta subunit 3 (POLD3), a protein functioning in early time points of HR, led to increased editing efficiency through rapid stimulation of HR without increases in off-target activity [45]. The mechanism of action of this fusion is based on quick accessibility to the DSB cut site through the rapid removal of Cas9 protein from the site by POLD3. The study provides a comparative analysis of editing efficiencies of Cas9-POLD3 to previously published tools, including Cas9-GE, Cas9-CtIP and Cas9-DN1S, targeting the *GFP* locus in the HEK293T cell model. Cas9-GE yielded the highest level of GFP expression; however, a comparison between the level of efficiency of HR and NHEJ at several endogenous loci in different cell models indicated a considerable disparity in HR efficiency, similar to what is reported by most knock-in studies. This study proved that while modest manipulation of editing efficiency is possible through such fusion models, crafting a universal Cas9 fusion that is similarly effective in various cell types is proving to be difficult. Thus, considering all the aforementioned reports, we conclude that MiCas9 provides superior HR enhancement, as it improves HR at levels higher than Cas9-GE, has reduced mutagenic potential, and shows consistency in multiple cell types and gene loci [43]. Curiously, despite the great promise, a survey of recent literature indicates that MiCas9 is only referenced in a single paper as of August 2023. This underscores the reluctance within the scientific community to shift from traditional Cas9 to embrace Cas9 variants that might be perceived as unsafe or insufficiently tested.

### 3.3. Epigenetic Modulation

The eukaryotic genome is tightly packed into a nucleoprotein structure called chromatin. The basic unit of chromatin is nucleosomes, which consist of 147 base pairs of DNA coiled around histone proteins [63] (Figure 2). The histone protein complex is formed by an octamer structure consisting of two units of four core proteins (H2A, H2B, H3 and H4) collated to make an array of nucleosomes that forms the dynamic chromatin architecture which may remain in an open (euchromatin) or closed (heterochromatin) conformation. Chromatin condensation is mediated by linker histones (H1) and epigenetic marks that control their higher-order closure. Epigenetic marks are post-translational modifications of histone tails, including but not limited to methylation, acetylation, phosphorylation, and ubiquitination. Collectively, these modifications form the epigenome code and substantially impact chromosomal topology. Such DNA packaging and chromatin architecture play a dynamic role in gene expression regulation by controlling the binding of transcriptional machinery. In fact, heterochromatin structure directly hinders access of RNA polymerase machinery and transcription factors to target genes. Thus, active gene expression requires the removal of nucleosome blockade to form non-compacted euchromatin, but how this impacts HR remains largely unclear at present. A range of different factors can initiate the transcriptional process [64,65], but pioneer transcription factors and chromatin remodelling complexes are known to play essential roles in initiating such chromatin rearrangements [66]. The fusion of pioneer factor p300 to dCas9 has been shown to greatly promote gene activation [67]. We speculate that the fusion of essential domains of pioneer factors to Cas9 may increase the capability of Cas9 to bind to the ‘closed’ chromatin domain and thus dramatically increase the efficiency of gene editing, particularly for genes located in heterochromatin locations. In addition, long noncoding RNAs have been reported to affect epigenetic regulators [68] and are important candidates to consider for future strategies aimed at enhancing HR through modulating chromatin states and histone modifications. 

The local chromatin environment around DNA breaks is thought to have an influence over HR efficiency, although very little is known about this fundamental area of molecular science. A plethora of studies have shown that dCas9 is directly fused to epigenetic modifiers, which modify histone post-translational modifications, facilitates specific gene activation [67,69] and downregulation [70]. While modulation of chromatin accessibility and nucleosome occupancy through epigenome editing can clearly impact gene expression, its effect on gene editing remains poorly understood. Ding et al. [47] attempted to improve precise editing by impacting chromatin structure by fusing Cas9 to chromatin modulators. The fusion of human HMGB1, HMGN1 and Histone 1 globular domain (H1) to Cas9 increased editing activity by 1.7–2.5-fold compared to unfused controls. A synergistic effect of double fusion of Cas9 with these proteins increased activity by 2.5–3.4-fold. The effect of doubly fused Cas9 to two different combinations, Cas9-HMGB1-HMGN1 and Cas9-HMGN1-H1, showed twice the efficiency in genome insertion compared to unfused Cas9. It is, however, difficult to discern whether this increased HR activity was due to increased chromosomal accessibility of Cas9 or enhancement of DNA repair machinery. Cas9-HMGB1-HMGN1 had a comparable HR to indel ratio to Cas9, whereas Cas9-HMGN1-H1 favoured HR over NHEJ with ratios of 31.8% HR to 20.7% indels. However, the reasons behind and the mechanisms through which different combinations of these peptides change HR efficiency are not well understood. Fusions of Cas9 systems with proteins that alter chromatin accessibility can potentially broaden the scope and safety of genome editing technologies, but a more comprehensive understanding of the interplay between chromatin dynamics and the HR and NHEJ repair machinery is essential to developing more precise editing tools.

Cas9 fusion proteins of histone epigenetic modifiers, such as methyltransferase (PRDM9), leverage modifications, such as trimethylation at H3K4 and H3K36 histone residues, to promote HR at DSBs [46]. These histone marks were previously shown to influence HR over NHEJ [71,72]. In there, Chen et al. revealed its site-specific action and demonstrated that chromosomal accessibility and epigenetic modification can play essential roles in influencing DNA repair pathways and thus affect the gene editing efficiency of CRISPR systems. However, the mechanism behind its impact on precise editing remains largely unresolved.

### 3.4. Colocalisation of Donor Sequence

Donor availability at DNA breaks influences the DNA repair process. An increase in the local availability of exogenous DNA in the nucleoplasm increases HR without tampering with the activity of proteins involved in break repair processes. One such strategy involves the fusion of avidin to the C-terminal of Cas9 with a flexible amino acid linker and biotinylated single-stranded oligo (SSO)[48]. High-affinity interaction between avidin and biotin increases the accumulation of donors at the cut site and increases HR to indel ratios. Similarly, the fusion of SNAP-tag technology to SaCas9 (Cas9 derived from *Staphylococcus aureus*) enriches local donor concentration via covalent attachment with O6-benzylguanine (BG)-labelled oligo [49,73] and increases HR at variable rates across different cell models and gene targets. Although higher editing efficiencies are observed, the modification and purification of donor sequences are complex and limit scalability and cost-effectiveness. Aird et al. tackled this limitation by fusing Porcine Circovirus 2 endonuclease Huh [74] to Cas9 to covalently link SSO via a recognition sequence of 13 base pairs at 5′ of the SSO [50]. This strategy is scalable and less complex compared to other donor localisation methods as this tethering does not require special donor design but utilises natural phospho-tyrosine bond formation between the Huh domain and recognition sequence in the oligos. Strikingly, RNP titration analysis revealed that lower concentrations of RNP increased HDR up to 15–30-fold by Cas9-Huh fusions compared to the unmodified Cas9 counterpart [50]. Another modified Cas9 with a similar mechanism of donor localisation as Cas9-Huh includes fusion with VirD2 relaxase, a virulence protein found in agrobacteria, which demonstrated a six-fold increase in the effectiveness of gene editing in plants [75]. This strategy has been applied to improve precise gene knock-in in rice [75] and highlights the importance of the development of Cas9 fusion variants for application in modern agricultural practices, such as the development of stress-tolerant crops. While donor saturation at the target site improved precise gene editing, most studies fail to report implications of such localisation of donors at off-target sites and rarely investigate the possibility of repeated insertions and other aberrant recombination events due to the abundant donor availability. Moreover, we presume excessive donor localisation may crowd the DSB site and inadvertently reduce DNA accessibility to cellular machinery and affect cell fitness, but no study has demonstrated this thus far. Other methods of accumulating donors at the target site through rational donor designs and fusion of gRNA have improved HR [56,76,77], but discussion on these initiatives is beyond the scope of this review as they do not involve direct fusion with Cas9. 

## 4. Structural Prediction of Fusions

Structural characterisation of Cas9-fusion proteins is required to understand the dynamics of the interaction of fused motifs with Cas9 and with DNA and nucleosomes for rational designing of optimised fusion proteins that elicit efficient editing. Here, we present Alphafold2 [78] predicted structural models of HR enhancers with proven activity in primary cells, as mentioned in the previous sections (Figure 3). Cas9 is predicted as a well-folded protein and aligns well with previous crystal structures. As expected, HR enhancers show limited secondary structure and are flexible relative to Cas9. To optimise computational efficiency, the Alphafold2 algorithm predicts structures at the lowest volume possible. As such, flexible sections of a protein are often modelled as in Figure 3, where they appear on first inspection to interact with more stable elements. However, closer inspection of the predicted aligned error (PAE) plots clearly demonstrates that these flexible regions are not interacting with Cas9. The relative orientation of the HR enhancers to Cas9 is highly flexible and would be, from a structural perspective, unlikely to adversely affect Cas9 function. Refining the structural modelling of fusion proteins and their anticipated ribonucleoprotein interaction sites would drive the improved design of engineered linkers that can promote optimal spatial orientation of the fused domains and thus significantly amplify HR enhancement. 

## 5. Safer Editing Alternatives

Although Cas9 endonucleases are guided by gRNA to specific target sites, there are several accounts of off-target activity, which remains a major concern in CRISPR technology [52]. Taking inspiration from the mechanism of action of dimeric ZFNs and TALENS containing cleavage domain of bipartite restriction endonuclease FokI [7,8], two studies demonstrated the fusion of dCas9 to FokI domain reduced off-target cleavage [5,79]. The cleavage activity of this tool is reliant on the engagement of two gRNAs in a proper, inverted orientation for simultaneous binding of two dCas9-FokI monomers at target sites to trigger DSB. The probability of unspecific binding of two adjacent dCas9-FokI fusion molecules to an off-target site is significantly lower than single molecule binding, which minimises the risk of off-target action. Similarly, dimeric nCas9 molecules nicking opposite strands of target sites also induce DSB with reduced off-target activity. nCas9 forms nicks with higher efficiency than FokI monomers but retains residual DSB-forming activity, so off-target effects persist [80]. Cas9-FokI has shown a 6.7-fold lower indel rate on average compared to nCas9 [79], proving this tool to be a safer option to generate DNA lesions. Gene corrective studies of phenylketonuria using this tool have provided proof of concept for its application in precision medicine [81]. However, the principle of Cas9-FokI is guided by various factors, including gRNA design and PAM sequences on opposing strands of the cut region with appropriate spacer distance, which limits the number of possible candidate genes. In addition, the introduction of large amounts of DNA to cells is required, which promotes cytotoxicity.

Microbial single-stranded annealing proteins (SSAP) fused to dCas9 offer a promising alternative for eliciting HR in a targeted region without any DNA breaks. Although the mechanism of action of the SSAP to mediate cleavage-free genomic integration of donor is not well understood, positive results of insertion at target loci with minimum off-target effects have been shown in HEK293T cells [82]. More work is required to validate the results in multiple cell lines and ex vivo models to address its potential as a new age precise gene editor without mutagenic potential. Using CRISPR fused to recombinases is another method of performing large genomic insertions without utilising DNA damage response [83]. However, a major constraint of this method is in the restriction to cells already edited with insertion landing sites. Such limitations make the generation of DSBs the most preferred method for gene knock-in, and efforts to improve DSBs, therefore, remain widely studied.

## 6. Discussion

CRISPR-Cas has enjoyed immense popularity and has been a highly followed topic ever since its discovery, with around 14,000 papers published on PubMed in just the past two years. Several biotechnology companies, such as Editas Medicine, Vertex Pharmaceuticals and Beam Therapeutics, have emerged as pioneers of commercialising CRISPR research, leading clinical trials and product commercialisation. According to the CRISPR Medicine News list, there are currently around 117 CRISPR-based active clinical trials, of which 61.5% target cancer (19 trials in phase 2) and 28% target blood disorders (4 trials in phase 3 and 15 in phase 2). Beyond cancers and haematological disorders, CRISPR technologies have been leveraged to study rare diseases and neurodevelopmental disorders for diagnostic purposes and to generate experimental models, such as engineered animal models and organoids [84,85,86,87]. Recently, the SNIPR biome has received immense success in the development of a CRISPR-Cas3-based microbial gene therapy (SNIPR001) which reduces E. coli burden in the gut, thereby decreasing the chances of fatal infections in patients fighting haematological cancers [88]. With hundreds of new companies joining the precision medicine market, there is rapid and continuous progress in the widespread application of this biotechnology. These innovations are increasingly focusing on novel ways to repurpose CRISPR technology for broader applications, as well as tackling gene editing limitations. According to a CRISPR market analysis report by Grand View Research (Report ID: GVR-1-68038-375-1), the global market is currently valued at USD 3.15 billion, with a forecasted annual growth rate of 17.15% to trend throughout the decade. This year, the first CRISPR therapy received the spotlight for obtaining regulatory approval as a potential cure for sickle cell disease and beta-thalassemia [89,90]. However, the promise of precise gene editing as a universal cure for gene-related diseases remain yet to be fulfilled. Translation to disease treatments includes major challenges, including editing efficiency, delivery methods, off-target effects, and immunogenicity [91]. While it is safe to say that CRISPR-based gene editing has met tremendous success in lab settings, a very limited section of research has been translated to the clinical trial phase with some unfortunate discontinuations along the way [92]. 

CRISPR does not follow the one-size-fits-all model like traditional medicine and is mainly focused on patient-specific care, especially in conjunction with CAR-T cell therapy [93]. This largely limits its clinical translation and widespread applicability in healthcare for the time being. Over the years, many different Cas9 variants and fusion protein models have been engineered to improve precise editing for gene replacement, as discussed in this review. However, these improved molecular scissors have not been widely embraced by the biotechnology community and classical Cas9 is still widely used despite its demonstrated shortcomings in efficiency and accuracy. Using antibiotic enrichment and sorting out edited cells from the population pool using selectable markers is a common practice in most studies, yet the knock-in efficiency remains low. Cas9 fusion proteins are a promising solution that can improve editing precision, but more extensive knock-in studies focused on editing efficiency, safety, biocompatibility, and precision are required to progress to in vivo preclinical and clinical studies. It is of the utmost importance to study nuanced differences in DNA repair pathways across various cell types and identify key regulators that can bring about desired improvement for homology-driven gene insertion. Improved HR strategies will not only benefit existing CRISPR therapeutics in development but also help find disease solutions for complex monogenic diseases with multiple SNP variants, genetic diseases caused by deletion mutations and polygenic diseases. For instance, the main variant of mutation in most patients suffering from cystic fibrosis is caused by the deletion of phenylalanine at the 508th position of the CFTR protein. Current modulators do not provide a permanent cure for the disease, which can be provided by gene replacement therapy by insertion of the correct CFTR gene [94]. Treatment requires improved methods of gene insertions, which can be provided by Cas9 fusion variants. HR enhancement strategies have the potential to emerge and lead corrective gene therapies for other serious syndromes such as DiGeorge, Cri du chat, Wolf-Hirschhorn, Netherton syndromes and more, which require large genomic insertions. Moreover, these Cas9 fusion variants can have a wide range of other applications, for example, to generate indel-free human embryonic stem cell lines for these disease conditions with heterozygous and homozygous mutations. In this review, we propose other possible fusion candidates, such as enzymes involved in posttranslational modification of HR regulators, pioneer factors, and noncoding RNAs, which have not been previously considered but have the capability to modulate the epigenome and Cas9 activity to favour HR at DNA lesions. Alternative avenues can also be found in other DSB repair regulators, including proteins involved in the Fanconi Anaemia pathway [95]. In addition, the use of protein tethering arrays, such as SunTag, MoonTag and SpyCatcher systems [20,96,97] for recruitment of multiple known HR effector proteins at target sites also warrants consideration. Preliminary studies have shown that epigenome editing can improve gene regulation, but its potential use in HR management is largely unexplored. As the field advances its knowledge of the protein structures, kinetics and molecular processes that direct HR, and with advances in the three-dimensional modelling of complex multi-protein fusion complexes that we touch upon in this review, are each key advances that will enable the rational design of safe, efficient and clinic-ready genome modification technologies in the future.

## Figures and Tables

**Figure 1 ijms-24-14701-f001:**
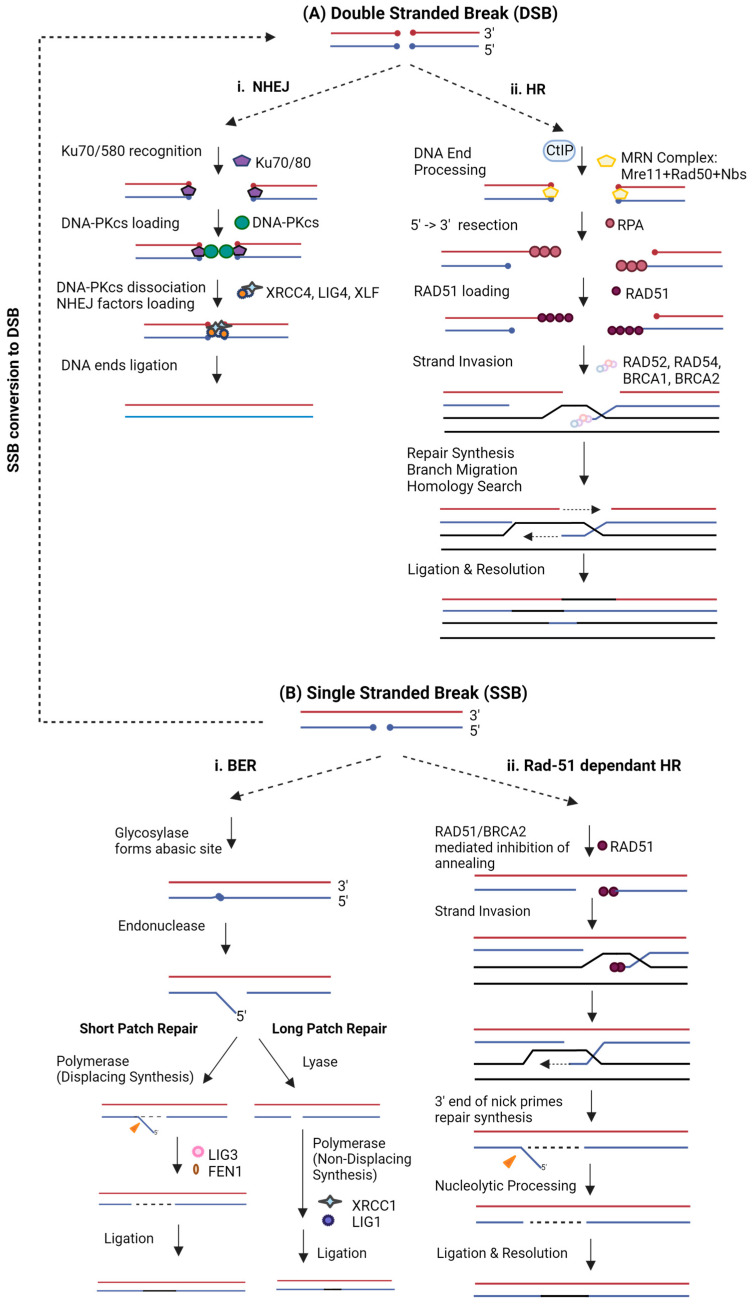
Schematic representation of the pathways of (**A**) double-strand break and (**B**) single-strand break repair processes, including key effectors involved in the regulation of each step.

**Figure 2 ijms-24-14701-f002:**
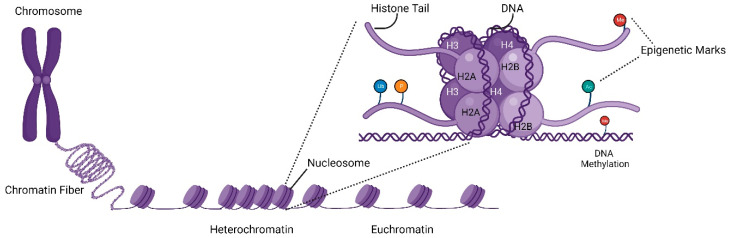
Chromatin barriers that impact Cas9 fusion protein-mediated HR.

**Figure 3 ijms-24-14701-f003:**
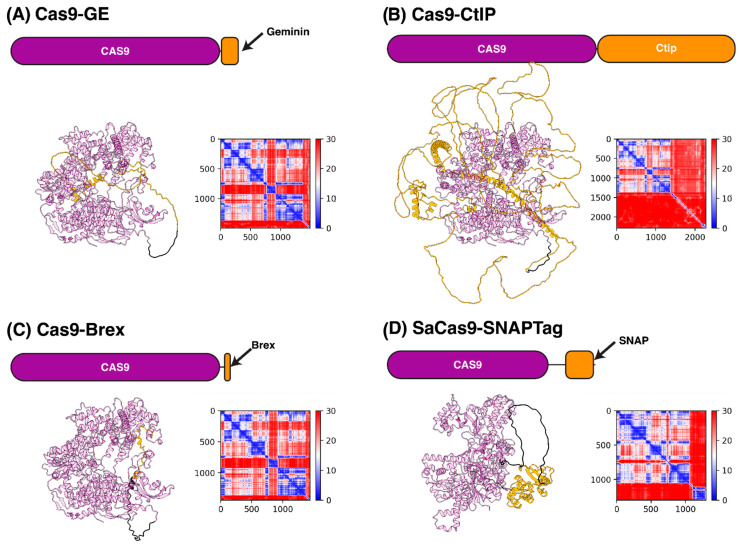
Structural prediction of Cas9-fusion proteins. For each fusion protein (**A**–**D**), we show (i) a cartoon representation of the size of the Cas9 protein relative to the HR enhancer and linker region. (ii) The top ranked Alphafold2 prediction of Cas9 (purple) HR enhancer (orange) fusion with the linker region (black). (iii) A predicted aligned error plot, a heat map that shows the correlation between any given residue in angstroms. Blue indicates that two residues are highly correlated with one another, while red indicates there is no correlation.

**Table 1 ijms-24-14701-t001:** Cas9 fusion proteins with proven effect in HR enhancement.

Factors	Fusion Component	Cell Models	HR Rate Fold Increase (x)	Mechanism of Action	Reference
A. Cell state	Geminin	HEK293T	1.5	Optimise timing of editing by Cas9 degradation in the G1 phase	[35,36,37]
		HiPSC	1.75		
		Foetal Fibroblasts	1.9		
	DNS1	HEK293T	1.5–3	NHEJ pathway inhibition at the DSB site	[38]
		K562	1.3		
		Jurkat Cells	1.1		
		LCL	2		
B. Repair factors	CtIP	Human fibroblasts	2	Increase colocalisation of HR proteins at the break site, promoting HR activity	[39]
		HiPSCs	1.5		
	UL12	HEK293FT	2		[40]
	RPA, MRN	HEK293-TLR	2.7		[41]
	Rad51	HEK293T	2.4		[42]
	Brex27	Fibroblasts	2.5		[43]
		AECs	2		
		iPSCs	2.1		
		Jurkat Cells	3		
	Exo1	A549	2–2.5		[44]
		K562	2–2.5		
	POLD3	HEK293T	1.4		[45]
		RPE-1	2		
		BJ-5ta	2		
C. Epigenetic state	PRDM9	HEK293T	1.6	Site-specific methylation of histone tail	[46]
	HMGB1	K562	1.7–2.5	Enhance chromatin accessibility	[47]
D. Donor	Avidin-Biotin	HEK293	2.5	Donor saturation at break site to increase HR frequency	[48]
	SNAP-Tag	HEK293T	3–24		[49]
		K562	17		
		mESC	2–6		
	PCV HUH endonuclease	HEK293T	2–3		[50]
		U2-OS	2–5		

## Data Availability

Data sharing not applicable.

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
