# Peer review of "CRISPR-Cas9 Direct Fusions for Improved Genome Editing via Enhanced Homologous Recombination"

_ijms, 2023, doi:10.3390/ijms241914701_

Round 1
Reviewer 1 Report
In this review article, the Authors discuss how fusion of Cas9 enzymes with the functional domains of various proteins that directly or indirectly impact DNA repair can enhance genome editing, favouring HDR over NHEJ. They provide pertinent, well explained examples of different Cas9 fusions, also including examples of fusions between defective/deadCas9 and functional protein domains.
Overall the article is well written with informative examples of different Cas9 fusion approaches, accompanied by relevant and objective discussion about the pros and cons of each approach and in some cases, their potential for clinical application. Figure 3 is a particularly nice inclusion in the article, and shows predicted structural models of HR enhancers fused to Cas9, as predicted by the Alphafold2 algorithm.
The minor corrections I suggest are:
to change the title to one that better reflects the focus on genome editing and enhancing HDR in Cas9 editing approaches.
on page 2, to correct "NHN-mutant" to "HNH-mutant".
Although it is implied and understood, the Authors should clearly state in the Introduction that the Cas9 referred to throughout the review is the most widely used one derived from Streptococcus Pyogenes.
on page 8, to define abbreviation of SpyCas9 as Streptococcus Pyogenes at first use since this abbreviation is not used anywhere else in the article. I assume that here the Cas9 strain is stated because the referenced work also relates to other Cas9 nucleases.
I would recommend that this article is accepted for publication.
Author Response
- change the title to one that better reflects the focus on genome editing and enhancing HDR in Cas9 editing approaches.
We have changed the title to "CRISPR-Cas9 direct fusions for improved genome editing via enhanced homologous recombination" as suggested.
2. on page 2, to correct "NHN-mutant" to "HNH-mutant".
Corrected, great pick-up.
3. Although it is implied and understood, the Authors should clearly state in the Introduction that the Cas9 referred to throughout the review is the most widely used one derived from Streptococcus Pyogenes.
We have now inserted "CRISPR-Cas9 derived from Streptococcus Pyogenes (referred to as Cas9 from here onwards) " on page 2, as suggested.
4. on page 8, to define abbreviation of SpyCas9 as Streptococcus Pyogenes at first use since this abbreviation is not used anywhere else in the article. I assume that here the Cas9 strain is stated because the referenced work also relates to other Cas9 nucleases.
We believe this is now clarified by defining cas9 as in point 3).
We thank the reviewer for the suggestions and careful reading of our manuscript.
Reviewer 2 Report
In this review manuscript, the authors review the progress made in Cas9-mediated Knock-In strategies toward genome editing. They focus on fusions on Cas9 effectors with various modalities that improve editing efficiency. The authors go into greater depth describing the mechanisms involved and discuss the utilization of molecules involved in repair pathways that can enhance efficient editing and reduce off-target effects and indel frequencies. Lastly, they discuss how computational modeling and AI tools such as alphafold2, can aid in protein engineering for optimal results.
Overall, the review is clear and adherent to the interest of the journal.
Below are my suggestions towards increasing the quality of the article for a broad readership:
1. It could be valuable to mention the biological and translational implications of these precise knock-in strategy. What are the application potential? Why the field would benefit from these developments of new knock-in strategy. Could the authors discuss and propose some application potential for engineering disease-relevant mutation, or correcting diseases/mutations in model systems? I suggest the authors study these published work and expand more on the application potential of various knock-in strategies in the discussion. (Wilde, Jonathan J., et al. "Efficient embryonic homozygous gene conversion via RAD51-enhanced interhomolog repair." Cell 184.12 (2021): 3267-3280.); Megagiannis, Platon., et al. “Reversibility and Therapeutic Development for Neurodevelopmental Disorders, Insights from Genetic Animal Models” Advanced Drug Delivery Review (2022).
2. It would also be informative to describe strategies for gene editing that introduce small scale insertions or replacements in the genome such as prime editing.
3. A discussion of the practicality and feasibility of the strategies mentioned in the paper, in terms of their translatability in diseases and their potential, would be enlightening. Mentions of some relevant disease applications would be useful to readers.
4. As a minor comment, some grammatical errors throughout the paper could be corrected.
Author Response
- It could be valuable to mention the biological and translational implications of these precise knock-in strategy. What are the application potential. Why the field would benefit from these developments of new knock-in strategy. Could the authors discuss and propose some application potential for engineering disease-relevant mutation, or correcting diseases/mutations in model systems? To address this suggestion we have now inserted additional text: "Another modified Cas9 with similar mechanism of donor localisation as Cas9-Huh includes fusion with VirD2 relaxase, a virulence protein found in agrobacteria, which demonstrated 6-fold increase in the effectiveness of gene editing in plants [75]. This strategy has been applied to improved precise gene knock-in in rice [75]and highlights the importance of development of Cas9 fusion variants for application in modern agricultural practices such as the development of stress-tolerant crops. " , "Beyond cancers and hematological disorders, CRISPR technologies have been leveraged to study rare diseases, neurodevelopmental disorders, for diagnostic purposes, and to generate experimental models, such as engineered animal models and organoids [85-88]. Recently, SNIPR biome has received immense success in development of a CRISPR-Cas3 based microbial gene therapy (SNIPR001) which reduces E. coli burden in gut, thereby decreasing chances of fatal infections in patients fighting hematological cancers[89]. " . This year the first CRISPR therapy received the spotlight for obtaining regulatory approval as a potential cure for sickle cell disease and beta-thalassemia [90, 91]. and " more extensive knock-in studies focused on editing efficiency, safety, biocompatibility, and precision are required to progress to in vivo preclinical and clinical studies. It is of the utmost importance to study nuanced differences in DNA repair pathways across various cell types and identify key regulators that can bring about desired improvement for homology driven gene insertion. Improved HR strategies will not only benefit existing CRISPR therapeutics in development but also help find disease solutions for complex monogenic diseases with multiple SNP variants, genetic diseases caused by deletion mutations and polygenic diseases. For instance, the main variant of mutation in most patients suffering from cystic fibrosis is caused by deletion of phenylalanine at the 508th position of the CFTR protein. Current modulators do not provide a permanent cure to the disease which can be provided by gene replacement therapy by insertion of the correct CFTR gene [95]. Treatment requires improved methods of gene insertions which can be provided by Cas9 fusion variants. HR enhancement strategies have the potential to emerge and lead corrective gene therapies for other serious syndromes such as DiGeorge, Cri du chat, Wolf-Hirschhorn, Netherton syndromes and more, which require large genomic insertions. Moreover, these Cas9 fusion variants can have a wide range of other applications, for example to generate indel-free human embryonic stem cell lines for these disease conditions with heterozygous and homozygous mutations."
- I suggest the authors study these published work and expand more on the application potential of various knock-in strategies in the discussion. (Wilde, Jonathan J., et al. "Efficient embryonic homozygous gene conversion via RAD51-enhanced interhomolog repair." Cell 184.12 (2021): 3267-3280.);Megagiannis, Platon., et al. “Reversibility and Therapeutic Development for Neurodevelopmental Disorders, Insights from Genetic Animal Models” Advanced Drug Delivery Review (2022). We have now inserted these references and discussed these as follows: "). Enriching Rad51 concentration in DNA repair microenvironment can promote homology directed repair as demonstrated by Song et. al. by use of RS-1, chemical agonist of Rad51 [61]. " and "
A recent study elucidated Rad51 localization at DSB site has applicability in homozygous gene conversion, doubling homozygous gene insertion efficiency in mouse models compared to Cas9 control [62]. This is of particular interest in improving gene corrective therapies for several genetic disease conditions as this repair system utilizes its own wildtype allele for homozygous knock-in bypassing the need for an exogenous template. This study provided a proof of concept for safer and efficient editing via Cas9 fusions by demonstrating that Rad51 localization can significantly enhance interhomolog repair in the mouse embryo."
- It would also be informative to describe strategies for gene editing that introduce small scale insertions or replacements in the genome such as prime editing. While prime editing involves cas9-fusion to a modifies reverse transcriptase it does not involve homologous recombination mediated repair and we therefore deemed that outside the scope of this review.
- A discussion of the practicality and feasibility of the strategies mentioned in the paper, in terms of their translatability in diseases and their potential, would be enlightening. Mentions of some relevant disease applications would be useful to readers. We believe we have now done this with the additional text outlined under point 1)